# Sleep Quality, Nutrient Intake, and Social Development Index Predict Metabolic Syndrome in the Tlalpan 2020 Cohort: A Machine Learning and Synthetic Data Study

**DOI:** 10.3390/nu16050612

**Published:** 2024-02-23

**Authors:** Guadalupe Gutiérrez-Esparza, Mireya Martinez-Garcia, Tania Ramírez-delReal, Lucero Elizabeth Groves-Miralrio, Manlio F. Marquez, Tomás Pulido, Luis M. Amezcua-Guerra, Enrique Hernández-Lemus

**Affiliations:** 1Researcher for Mexico CONAHCYT, National Council of Humanities, Sciences and Technologies, Mexico City 08400, Mexico; 2Clinical Research, National Institute of Cardiology ‘Ignacio Chávez’, Mexico City 14080, Mexico; 3Department of Immunology, National Institute of Cardiology ‘Ignacio Chávez’, Mexico City 14080, Mexico; mireya.martinez@cardiologia.org.mx (M.M.-G.); lmamezcuag@gmail.com (L.M.A.-G.); 4Center for Research in Geospatial Information Sciences, Aguascalientes 20313, Mexico; tramirez@centrogeo.edu.mx; 5Department of Electrocardiology, National Institute of Cardiology ‘Ignacio Chavez’, Mexico City 14080, Mexico; 6Cardiopulmonary Department, National Institute of Cardiology ‘Ignacio Chávez’, Mexico City 14080, Mexico; tomas.pulido@cardiologia.org.mx; 7Computational Genomics Division, National Institute of Genomic Medicine, Mexico City 14610, Mexico; 8Center for Complexity Sciences, Universidad Nacional Autónoma de Mexico, Mexico City 04510, Mexico

**Keywords:** poor quality sleep, social development index, nutrients, machine learning, features selection, balancing methods, Mexico City, Tlalpan 2020 cohort

## Abstract

This study investigated the relationship between Metabolic Syndrome (MetS), sleep disorders, the consumption of some nutrients, and social development factors, focusing on gender differences in an unbalanced dataset from a Mexico City cohort. We used data balancing techniques like SMOTE and ADASYN after employing machine learning models like random forest and RPART to predict MetS. Random forest excelled, achieving significant, balanced accuracy, indicating its robustness in predicting MetS and achieving a balanced accuracy of approximately 87%. Key predictors for men included body mass index and family history of gout, while waist circumference and glucose levels were most significant for women. In relation to diet, sleep quality, and social development, metabolic syndrome in men was associated with high lactose and carbohydrate intake, educational lag, living with a partner without marrying, and lack of durable goods, whereas in women, best predictors in these dimensions include protein, fructose, and cholesterol intake, copper metabolites, snoring, sobbing, drowsiness, sanitary adequacy, and anxiety. These findings underscore the need for personalized approaches in managing MetS and point to a promising direction for future research into the interplay between social factors, sleep disorders, and metabolic health, which mainly depend on nutrient consumption by region.

## 1. Introduction

Metabolic Syndrome (MetS) is a condition that increases the risk of developing or worsening several serious health conditions such as diabetes, heart disease, and stroke, as well as cognitive decline and dementia [1]. Sleep disturbances such as insomnia, apnea, and snoring, linked to MetS, can exacerbate these health risks [2,3]. In 2017, the National Health and Nutrition Survey of Mexico [4] estimated the prevalence of sleep disorders in Mexicans using a sample of 8649 people older than 18 years old. The results showed a prevalence of snoring while sleeping of 48.5%, difficulty sleeping of 36.9%, and tiredness or fatigue during the day of 32.4%; likewise, insomnia was 18.8% more prevalent in women. Regarding apnea, the results indicated that 23.7% had a higher risk of presenting apnea, especially the populations of those who were overweight and obese, hypertensive, and those over 40 years of age. In another study [5], the prevalence of insomnia was 36.7%, being more common among women (with a prevalence of 41.9%) than men (with a prevalence of 36.7%). Effective treatment for sleep disorders hinges on identifying their specific type and underlying causes, highlighting the ongoing need for improved diagnosis and treatment strategies.

The prevalence data on sleep disorders underscore the importance of understanding their impact on conditions like MetS. This underscores the necessity of employing tools such as the Medical Outcomes Study Sleep Scale (MOS) [6] in research to assess sleep quality and its influence on health. Its widespread use in diverse research studies [7,8,9] has deepened studies of how sleep disorders affect various health conditions and populations, thanks to its ability to measure multiple sleep-related aspects.

Similarly, nutrition and specific nutrients play crucial roles in developing and managing MetS [10]. MetS is a cluster of conditions that includes abdominal obesity, insulin resistance, dyslipidemia, and hypertension. Poor dietary choices and other lifestyle factors can contribute to developing and exacerbating these risk factors [11,12]. Excessive caloric intake, especially from high-fat and high-sugar diets, contributes to obesity; in consequence, it can contribute to insulin resistance, which is a key feature of metabolic syndrome. Low consumption of dietary fiber, commonly found in fruits, vegetables, and whole grains, is associated with insulin resistance. Diets high in saturated and trans fats can lead to dyslipidemia, which is characterized by elevated levels of triglycerides and low-density lipoprotein cholesterol and decreased high-density lipoprotein cholesterol. This lipid profile is a risk factor for cardiovascular diseases associated with metabolic syndrome. In contrast, omega-3 fatty acids, found in fatty fish, flax seeds, and walnuts, have been associated with favorable lipid profiles and may have a protective effect against metabolic syndrome [13,14,15,16]. As expected, nutrition and dietary habits are associated with MetS; various research has found the contributions of nutrients through applying diverse statistical models on the increasing or decreasing risk [17,18,19].

In the same way, another factor significantly associated with MetS is the social development index (SDI) [20], which is a composite measure of social and economic development. The SDI serves as a metric to evaluate the well-being and social progress in Mexico. Originating in the early 2000s and modeled after the Human Development Index (HDI), the SDI categorizes the level of social development in territorial units. These units correspond, for instance, to the subdivision of municipal geostatistical areas in Mexico City. The SDI employs a methodology established by the National Council for the Evaluation of Social Development Policy (Consejo Nacional de Evaluación de la Política de Desarrollo Social, CONEVAL) for its calculation (refer to Methods for further details on the SDI) [21].

Countries with higher SDI scores tend to have better health outcomes, including lower rates of MetS [22], and an additional study connects the risk of MetS with economic and social vulnerability as well as inappropriate nutrition profiles [23]. The evidence suggests a close association between the SDI and sleep disturbances, which is a relationship influenced by socioeconomic factors such as income level and education. These factors directly affect access to health services and lifestyle habits, such as diet and physical activity, which are essential for maintaining optimal sleep quality. Analyzing how the SDI and sleep disturbances interact with MetS is crucial to unravel the social and economic determinants that shape these complex interconnections. Understanding these dynamics will not only facilitate the identification of the types of sleep disorders that increase the prevalence of MetS but will also contribute to developing more effective strategies for its prevention and treatment, thus improving overall health and well-being. For this reason, developing automated methods for diagnosing sleep disorders, identifying the determinants of the SDI, and predicting MetS have become fields of significant research interest.

In the case of sleep disruption, machine learning has shown promise in improving the accuracy and efficiency of the diagnosis process. The work of Mencar et al. [24] presents the application of five machine learning models to predict the severity of obstructive sleep apnea syndrome (OSAS) using polysomnography data, where the random forest model obtained the highest accuracy (90.91%) and relevant features such as respiratory rate and oxygen saturation were extracted. Another study [25] applies a machine learning model to predict the presence of OSAS using clinical and demographic data. The random forest model performed best, achieving an accuracy of 87.1%. The most important predictors were body mass index (BMI), age, and gender, as well as additional predictors such as neck circumference and smoking.

In another study by Eyvazlou et al. [26], an ANN model was developed to predict MetS based on sleep quality and work-related risk factors. The results showed that the ANN model could identify individuals at risk of MetS with a sensitivity of 74.1% and a specificity of 76.2%. Moreover, other studies [27,28] have also applied machine learning to understand the social determinants that affect and influence the health of individuals.

However, despite the excellent results described in previous studies, one of the most common challenges in medical diagnoses is the issue of class imbalance. This problem significantly impacts the performance of classifiers, as they tend to exhibit a bias towards the majority class, resulting in skewed outcomes. In this context, authors such as Kim et al. [29] propose a prediction model that utilizes balancing techniques to identify middle-aged Korean individuals at a high risk of MetS. The dataset used in their study comprises age, gender, anthropometric data, sleep quality, and blood indicators of 1991 individuals. The results showed that XGBoost (using Scikit-learn library in Python ver. 3.8.5), employing SMOTE, achieved an AUC of 85.1%.

The present study examines the connection between the SDI, sleep disturbances, types of nutrients consumed, and MetS within a cohort from Mexico City. We aim to identify critical factors that may be key to reducing MetS incidence or severity by applying machine learning algorithms. Additionally, we will use data balancing techniques to improve the predictive performance of our models and enhance feature selection. By incorporating these methods, we aim to uncover valuable insights and contribute to developing more accurate and practical approaches for addressing MetS.

## 2. Materials and Methods

### 2.1. Data

Data for this study were derived from the baseline assessment of a cohort called Tlalpan 2020 from the National Institute of Cardiology Ignacio Chávez in Mexico City [30]. This project was authorized by the Institutional Ethics Committee of the National Institute of Cardiology Ignacio Chavez under code 13-802. The dataset used in this investigation includes data from 3156 volunteers (all of them were informed of the research purposes and signed a letter of informed consent) about their anthropometric measurements, consumption of alcohol and tobacco, level of physical activity, level of economic income, level of education, anxiety, family history health, biomedical evaluation, quality of sleep, and the amount of nutrients consumed.

#### 2.1.1. Quality of Sleep

The sleep quality was measured using MOS [6], a self-report for assessing sleep quality and quantity. This questionnaire includes 12 items about sleep disruption, snoring, sleep shortness of breath or headache, sleep adequacy, and sleep somnolence; it additionally measures the number of hours of sleep per day over the previous four weeks. The MOS has been used in several studies, such as discriminating the quality of sleep among a Spanish postmenopausal population [9], diagnosing cases of apnea [7,8], and identifying sleep disturbance in patients with rheumatoid arthritis [31], among others.

#### 2.1.2. Clinical and Anthropometric Parameters

Clinical and anthropometric data such as systolic blood pressure (SBP) and diastolic blood pressure (DBP) (measured according to standard procedure [32]) were collected, as well as waist circumference (WC), height and weight (measured according to ISAK [33]) for calculationof BMI, and the height–waist index (WHtR). These were calculated from primary measurement data.

#### 2.1.3. Biochemical Evaluation

The following laboratory test measurements corresponding to blood samples were included: glucose (GLU), triglycerides (TRIG), HDL cholesterol (HDL), LDL cholesterol (LDL), uric acid (URIC), atherogenic index (IAT), and sodium (NA).

#### 2.1.4. Social Development Index

Comprising key dimensions associated with education, health, and housing, the SDI incorporates specific indicators for the evaluation of each dimension. The weight assigned to each indicator varies based on its significance in the overall assessment of social development. The resulting scores are aggregated to yield a score for each dimension. The SDI value facilitates the ordering of territorial units based on their achieved levels of development, classified as Very Low, Low, Medium, and High [34,35].

The SDI indicators (as reported in reference [21]) are briefly described below:Quality and available space in the home (QUA_HOUS): The quality of housing is measured by the type of flooring, and the amount of living space is indicated by the number of people per bedroom, with two being the standard.Educational access (EDULAG): This indicator measures the proportion of people aged 18–59 who have completed secondary school or have received 13 years of schooling, which is considered a minimum standard for well-being.Access to social security and/or Medical Service (HEALTHAC): This indicator measures the coverage of any of the Mexican health systems.Durable goods (DURAB): This indicator measures possession of material goods whose value is equal to or greater than USD 17.81, or possession of at least three items such as a television, gas stove, computer, refrigerator, or washing machine.Sanitary adequacy (SANITRY): This indicator measures the availability of a water supply, toilet facilities, and access to a drainage system.Electricity access (ENER_AD): This indicator measures whether or not there is adequate access to electricity.

#### 2.1.5. Habits and Factors Associated with Lifestyle

Furthermore, habit data were also collected, such as habitual smoking, alcohol consumption, and physical activity (calculated based on the International Physical Activity Questionnaire, IPAQ, Ref. [36] by metabolic equivalent minutes/week, which are classified in the following categories: low, moderate, and high).

Education level was collected and classified into three categories: primary school, high school, and university studies, as well as postgraduate school. Similarly, we collected the level of economic income, which was classified into three categories based on the Mexican peso income paid monthly: low (MXN 1.00 to MXN 6600.00), medium (MXN 6601.00 to MXN MXN 11,000.00), and high (more than MXN11,000.00).

#### 2.1.6. Psychological Stress Level

We used the State-Trait Anxiety Inventory (STAI) to collect data about psychological stress levels, which were categorized into five categories: high (>65), moderate (56–65), medium (46–55), minor (36–45), and low (<35) [37,38].

#### 2.1.7. Dietary Information

To gather information about the frequency of food consumption and other dietary products, we utilized a software tool called the “Evaluation of Nutritional Habits and Nutrient Consumption System“ from the National Institute of Public Health [39]. This system examines the meals individuals have consumed over a day within the previous year and computes the quantity of nutrients ingested.

All data mentioned in this section are presented in the Table 1.

### 2.2. Methods

#### 2.2.1. Feature Selection

Feature selection is essential to identify and establish the most critical variables.

In this study, we employed logistic regression to measure the relationship between variables and class alongside machine learning algorithms to discern the most significant features. The algorithms used were RF and RPART (see Machine Learning Modelsbelow), applying the mean decrease accuracy for calculating variable importance, which can be expressed as follows:(1)$$MDI_{i}=\sum_{all\;nodes}\left(\left(Imp\left(node\right)-Weight.Imp\left(node\right)\right)/NS.N\right)$$
where $MDI_{i}$ is the mean decrease impurity of the ith variable; Imp(node) is the impurity of the node before the split; Weight.Imp(node) is the weighted impurity of the child nodes resulting from the split; and NS.N is the number of samples in the node before the split.

#### 2.2.2. Balancing Methods

Balancing methods such as SMOTE and ADASYN have helped address the class imbalance issue within our dataset.

ADASYN (Adaptive Synthetic Sampling), which is part of the UBL R package, takes a unique approach by generating synthetic samples based on the local density of minority class instances, with a focus on instances that are more challenging to learn. In this method, the β parameter controls the desired balance rate between the minority and majority classes during the generation of synthetic samples. When β is set to a value greater than 1, a proportionally larger number of synthetic samples will be generated relative to the instances of the minority class. This further increases the ratio between the minority and majority classes.

The second method, SMOTE (Synthetic Minority Oversampling Technique) of the performanceEstimation R package (Version: 1.1.0), generates synthetic samples for the minority class. In SMOTE, the *k* parameter determines the number of nearest neighbors used to generate synthetic samples. A small value of *k* can lead to an excessive generation of synthetic samples that may be too close together, resulting in model overfitting. Moreover, if *k* is too large, synthetic samples may be less representative of the minority class and fail to capture data variability adequately.

#### 2.2.3. Machine Learning Models

To build the models, we applied two machine learning algorithms, RF [41,42] and RPART [43,44], as well as PCA [45,46]. RF, introduced by Breiman [47], is a machine learning algorithm combining multiple decision trees to create a model with the highest accuracy. Rpart (Recursive Partitioning and Regression Trees), by Breiman [48], works by recursively partitioning the input data based on predictor variables to create a tree-like structure. This algorithm aims to find the optimal splits in the data that maximize the homogeneity or purity of the resulting subgroups. Principal component analysis (PCA) is a data analysis technique used to simplify the complexity of data by reducing their dimensionality, facilitating visualization and analysis.

### 2.3. Performance Measures

We used sensitivity, specificity, and balanced accuracy (B.ACC) to evaluate model performance. These metrics provide a fair assessment of the model’s performance across all classes, considering the issue of class imbalance.
(2)$$SENS=\frac{TP}{TP+FN}$$
(3)$$SPC=\frac{TN}{FP+TN}$$
(4)$$B.ACC=\left(\frac{1}{2}\right)\left(\frac{TP}{P}+\frac{TN}{N}\right)$$
where *P = Positive, N = Negative, TP = True Positive, FN = False Negative, TN = True Negative, and FP = False Positive*, respectively.

## 3. Statistical Analysis and Development of Prediction Models

All experiments were performed using the R programming language (3.6.1) [49]. Min-max was used to normalize continuous variables, and dichotomous variables were represented as numbers. Figure 1 provides a general overview of the experimental process described in this section. To develop predictive models, it was necessary to process the data and implement a balancing technique. The minority class was oversampled, taking into account the majority class. As a first step, SMOTE was applied, and it was necessary to determine the best value of *k* (number of nearest neighbors), so experiments were conducted by varying *k* (here, we present k=1, k=5, and k=9). In this process, the dataset was randomly divided into 70% for training and 30% for testing. To accomplish this task, we applied two machine learning algorithms, RF and RPART. In the case of RF, we varied the *mtry* parameter from 1 to 10 and considered *ntree* values of 100, 300, 500, and 1000 for each model.

Additionally, a subset of features was extracted in each created model using the variable importance (VarImp) of RF, and a 10-fold cross-validation was performed. Similarly, in the case of RPART, parameter tuning was conducted by considering cp=0, cp=0.05, and cp=0.005, using a 10-fold cross-validation. Likewise, a subset of features was extracted in each created model.

Once the feature subsets were obtained, along with the optimal value for each corresponding parameter of each algorithm and data balancing technique, we tested the generated feature subsets using RF and RPART. This was accomplished by conducting 30 runs with different seeds to assess the performance of each model. In all experiments, a minimum of 30 independent runs were conducted for each algorithm using 30 different seeds. The mean and standard deviation of the performance measures were calculated for each of these runs.

## 4. Results

Understanding how MetS, nutrition, sleep disturbances, and SDI relate in men and women can have important clinical and public health implications. In this study, we used logistic regression before dataset balancing to pinpoint the critical variables associated with MetS in both sexes. Table 2 presents the results of the features and their corresponding values obtained.

Analyzing the data, in men, the top 10 variables most related to MetS are GLU, TRIG, WC, IAT, SBP, vitamin B12 (B12), BMI, lactose (LACT), carbohydrates (CARBO), and high glucose levels based on the dietary survey (GLU_1). Conversely, in women, the ten most relevant variables include GLU, TRIG, WC, BMI, SBP, total proteins (PROTEI), fructose (FRUCT), high cholesterol total based on the dietary survey (CHOL_SN), URIC, and copper (CU). To achieve a more effective visualization of these prominent features from the logistic regression for both men and women, Figure 2 is presented. Red square symbols represent the most substantial variables for women, while blue triangles represent those for men. A cautionary note must be made for the seemingly outlier behavior of blood glucose and triglycerides with very high coefficients. Let us recall that these features are closely related to the very definition of MetS. Such variables were included in our models only for the sake of database completeness and comprehensiveness. Detailed results for women can be found in Appendix A, and those for men are available in Appendix A.

Subsequently, we employed SMOTE and ADASYN with RF and RPART to reassess the most influential features associated with MetS prediction within a now balanced dataset. Following this, with the data balancing techniques effectively applied and their parameters fine-tuned, we extract feature subsets by utilizing RPART and RF for both women and men. Extracting features related to MetS in a balanced dataset improves model generalization (conducting training more evenly and accurately), optimizing performance, and reducing overfitting. Considering the challenges associated with including all variables in a model, such as noise, redundancy, and overfitting, we extracted the 17 variables with the highest values obtained in each model of RF and RPART after applying SMOTE and ADASYN. 

The extracted feature subsets, along with their respective values, are presented in Table 3, Table 4, Table 5 and Table 6. These tables also detail the employed balancing technique for each set of variables and their corresponding parameters ranging from 1 to 5. Each subset was adjusted for its corresponding parameter, B for ADASYN and *k* for SMOTE, considering values of 1 and 5.

Similarly, Table 7 showcases the performance achieved by the RF algorithm, while Table 8 presents the performance of the RPART algorithm. In both tables, the Value column provides information regarding the relative importance of each feature.

### 4.1. Best Features for Men Using RF and ADASYN/SMOTE

Specifically, Table 3 exhibits four feature subsets obtained from male data using random forest with ADASYN and SMOTE. According to Table 7, the most effective subset was obtained by applying ADASYN with B = 1 with a balanced accuracy of 86.22% and a deviation standard of 0.26%.

The most influential factor within this subset was BMI, which had a significant importance value of 92.9499. This was followed by WEIGHT and energy efficiency (ENER_AD), with importance values of 49.4782 and 48.8887, respectively. Other factors such as educational lag (EDULAG), common-law marriage (LIV_TOG), durable goods (DURAB), and maternal gout history (MOTHERGT) also contributed to the model, albeit to a lesser extent.

### 4.2. Best Features for Men Using RPART and ADASYN/SMOTE

In the case of features obtained by RPART (see Table 4), using both SMOTE and ADASYN, the results were slightly worse than those obtained with RF (Table 3). In this scenario, the best subset was achieved by the subset with the parameter ADASYN = 5, which achieved an 82.32% balanced accuracy metric with a standard deviation of 0.99% (see Table 8).

Switching gears to the outcomes yielded by random forest with ADASYN using a B value of 5, BMI takes center stage with a substantial value of 683.74, signifying its paramount role in predicting the outcomes related to the examined condition. Following closely in significance are ENERGY_AD and EDULAG, boasting values of 619.99 and 565.33, respectively, both making substantial contributions to predictive capability. ALCOHOL and WEIGHT also exhibit noteworthy importance with values of 355.97 and 295.25, underlining their relevance within the model. Moreover, features like divorce (DIVORC), no academic degree (NONE), and MOTHERGT, while exerting a comparatively lower influence, still contribute to the model’s predictive capacity, as indicated by their respective values.

### 4.3. Best Features for Women Using RF and ADASYN/SMOTE

The random forest model using SMOTE with k=5 achieved the best performance for women, reaching an 88.50% accuracy with a standard deviation of 0.40% (see Table 7). In this case, Table 5 reveals that BMI was identified as the primary predictor, with a notable value of 484.31, clearly highlighting the critical importance of BMI in predicting MetS in this particular context. Additionally, IAT (481.48) and WEIGHT (339.17) also showed significant associations, further emphasizing the relevance of weight-related measurements.

Including sleep disturbances (SLPSNR1, SLPSOB1, BREATH, DROWSY, and SLPNOTQ) and even cholesterol levels (CHOL_ANT) among the influential variables underscores their pivotal contributions to MetS prediction in women. The importance of AGE and SDI parameters like sanitary adequacy (SANITRY) is also noteworthy. It is essential to highlight that psychological factors such as trait anxiety (TRAIT_ANX) were included, accounting for the potential influence of mental health aspects in MetS prediction.

### 4.4. Best Features for Women Using RPART and ADASYN/SMOTE

In this instance, SMOTE with k = 5, combined with RPART, achieved the best performance, attaining a balanced accuracy of 84.49% with a standard deviation of 1.43% (see Table 8). The results of the corresponding subset (RPART applied to women’s data using SMOTE with a parameter value, k=5) shown in Table 6 reveal that the most influential feature was IAT, with a value of 483.23, followed closely by BMI and WEIGHT, which have values of 410.37 and 409.78, respectively. Features like URIC, snores during sleep (SLPSNR1), somnolence (SLPS3), SODIUM, vitamin E consumption (VITE), and habitual smoking (SMOKING) also exhibit noticeable influence, indicating their relevance in understanding the targeted phenomenon. Conversely, some nutrients like sucrose (SUCR), maltose (MALT), and FRUCT have relatively lower values; however, they can provide valuable information about dietary habits, nutritional deficiencies, or behaviors related to MetS.

This study’s results, employing random forest and RPART algorithms and SMOTE and ADASYN techniques for both genders, offer valuable insights. These results underscore the importance of health and lifestyle elements in MetS prediction, encompassing sleep disturbances, cholesterol levels, age, psychological factors, and SDI parameters.

### 4.5. Analyzing the Best Features Using PCA

Based on the results of the features obtained in the best models, we used PCA to visually and graphically analyze the top features for men and women to explore potential correlations and latent patterns among these influential factors and reduce dimensionality to the greatest possible extent.

In the case of men, we considered feature subsets obtained from the random forest model using ADASYN with B = 1 and RPART with ADASYN and B = 5. The subsequent features were integrated: BMI, WEIGHT, ENER_AD, EDULAG, LIV_TOG, DURAB, MOTHERGT, IAT, HEALTHAC, DIVORC, QUA_HOUS, STRATUM, FATHERGT, NONE, MARRIED, VALUE, URIC, SANITRY, SINGLE, and ALCOHOL.

For women, we considered feature subsets obtained from the random forest model with SMOTE and k=5 and the RPART model with SMOTE and k=5. These models are regarded because they achieved the highest performance (see Table 7 and Table 8 where extremely small percentage uncertainty values in Table 8 are shown rounded down to 0.00 for clearer presentation). The following features were included: BMI, IAT, WEIGHT, URIC, SLPSNR1, CHOL_ANT, AGE, SLPSOB1, BREATH, TRAIT_ANX, SMO_PASS, SANITRY, MOTHERDL, DROWSY, SMOKING, SINGLE, EXSMOKER, SEC_SCHOOL, SLPNOTQ, SLPS3, SODIUM, ALCOHOL, SATFAT, MONFAT, NA, VITE, FATHERDB, SUCR, MARRIED, FRUCT, ZN, and MALT.

The PCA analysis, as shown in Figure 3, revealed the relative importance of features concerning MetS in men. The first principal component (PC1) was more influenced by features such as WEIGHT, BMI, and SDI by value (VALUE), suggesting that these variables significantly contributed to the observed variability in the data. On the other hand, the second principal component (PC2) was more affected by features like EDULAG and socioeconomic stratum (STRATUM). These findings indicated that weight and BMI were prominent factors in the context of MetS, as well as education and socioeconomic stratum. In this case, PC1 was considered the most significant component, as it had a magnitude of 0.508501, capturing most of the variability, while PC2 had a magnitude of 0.499809.

On the other hand, in the case of women (see Figure 4), features associated with the variability of MetS along PC1 were sodium levels based on the dietary survey (SODIUM), saturated fat (SATFAT), and monosaturated fat (MONFAT), which exhibit significant magnitudes in PC1. Furthermore, BMI significantly influences PC1, indicating its association with this variability. Conversely, variables like short sleep duration (SLPSOB1) and waking up with shortness of breath (BREATH) demonstrate significant magnitudes in PC2. Similarly, TRAIT_ANX and feeling drowsy or sleepy (DROWSY) are associated with variability in PC2. Therefore, considering the magnitudes in the principal components, the features in women associated with the risk of MetS include SODIUM, SATFAT, and MONFAT from PC1, as well as SLPNOTQ and SLPSOB1 from PC2.

## 5. Discussion

MetS is a severe and potentially life-threatening condition that significantly increases the risk of developing cardiovascular diseases and also increases the severity of diabetes. Over the years, several consistently highlighted risk factors have been associated with MetS. Considering imbalanced data, this study analyzed participant data from a cohort to identify the primary risk factors in both men and women. Subsequently, data balancing techniques were applied to ascertain whether significant differences exist, contributing to selecting risk factors for MetS prediction. Using data balancing techniques is crucial in this context, as it helps ensure a more accurate and unbiased identification of relevant risk factors, especially when working with unevenly distributed data. In this study, we applied logistic regression to identify the risk factors in men and women that predict the occurrence of MetS within an imbalanced data environment.

### 5.1. Logistic Regression

The logistic regression analysis in women demonstrates (as expected, of course) the strong connection between MetS and elevated glucose levels, which is in line with prior research [50,51], emphasizing the crucial role of glucose in MetS. Additionally, uric acid is also identified as a significant risk factor in women [52,53,54]. Subsequent findings revealed other risk factors, including waist circumference, BMI, and systolic blood pressure, which are all essential components of MetS. WC is an indicator of abdominal obesity closely linked to insulin resistance, while BMI reflects the relationship between weight and height, which is a significant obesity-related risk factor for MetS.

Furthermore, Figure 2 highlights additional significant factors derived from dietary data, including the intake of protein and fructose [55,56,57]. When these two nutrients are combined, they have been linked to an elevated risk of MetS [58]. Likewise, copper consumption is evident, which can impact glucose regulation [3] and liver function, which are both crucial components in MetS [59]. These factors underscore the importance of moderate consumption of these nutrients in preventing MetS.

In the case of men, glucose was identified as the primary factor associated with MetS, followed by triglycerides, waist circumference, atherogenic index, and systolic blood pressure. Additionally, the consumption of lactose [60] and carbohydrates [61] was noted among the nutrients. Elevated glucose, triglycerides, and waist circumference are critical markers of MetS, while the atherogenic index assesses cardiovascular risk. High systolic blood pressure is another significant component of this syndrome. Regarding lactose, it is worth noting that certain dairy products may include added sugars, which can potentially increase the overall calorie intake [62]. This potentially contributes to obesity and insulin resistance, which are two critical factors in the onset of MetS. Moreover, high lactose consumption is associated with a risk factor for developing diabetes, cardiovascular diseases, and increased cholesterol levels [63,64].

It is possible that when working with unbalanced datasets, machine learning models like logistic regression tend to be biased towards the majority class. For this reason, data balancing techniques such as SMOTE and ADASYN were used to enable a more equitable training of the models to identify more precise relationships between variables and the MetS.

### 5.2. Use of Machine Learning with Synthetic Data

The most effective machine learning models for women revealed associations with attributes related to sleep quality, such as snores during sleep [65], short sleep duration (SLPSOB1) [66], waking up with shortness of breath (BREATH) [67], restless sleep (SLPNOTQ) [68], and somnolence (SLPS3). Multiple studies have shown that poor sleep quality is closely linked to cardiovascular disease [69,70], diabetes [71], and MetS [72], as well as other adverse health outcomes. In the case of women, an increased likelihood of facing significant risks related to cardiovascular diseases and sleep problems has been observed, especially for those in the postmenopausal stage, which, in turn, can contribute to the development of risks associated with MetS [73]. Additionally, they highlighted factors related to anxiety (TRAIT_ANX), despite the association between MetS and anxiety remaining a subject of debate due to various issues [74], this study, like some others [75,76,77,78], identified anxiety as one of the critical factors that predisposing women to MetS.

In the same way, ex-smokers and current smokers (EXSMOKER, SMOKING) were found to be relevant features; based on this, it has been observed that both smokers and former smokers are predisposed to MetS. This finding is supported by various studies that suggest that smoking can have an adverse impact on blood lipid levels and lead to metabolic disturbances [79,80,81].

In women, nutritional components also appeared as relevant features, such as SATFAT, MONFAT, SUCR, FRUCT, and MALT. Based on this, a study has revealed that fructose, sucrose, and maltose are critical components of the leading nutrient pattern associated with a higher risk of MetS [58].

In the case of men, the most effective machine learning models displayed more pronounced associations with features linked to the SDI, encompassing ENER_AD, EDULAG, durable goods (DURAB, HEALTHHAC), quality and living space (QUA_HOUS), socioeconomic stratum (STRATUM), social development index by value (VALUE), and sanitary adequacy (SANITRY). In studies [22,82,83,84], a significant association has been observed between a low socioeconomic level and the prevalence of metabolic syndrome. Furthermore, these models underscored variables related to parental gout conditions (MOTHERGT, FATHERGT). This supports research exploring the genetic predisposition to gout and suggests that a family history of this disease may increase the risk of other family members developing it [85]. This condition may also be related to metabolic syndrome due to poor dietary habits that could lead to obesity and insulin resistance [86,87].

### 5.3. Principal Component Analysis

Based on the resulting features obtained for men and women via machine learning models, we applied principal component analysis to identify trends and potential correlations. The PCA conducted using the features obtained for men (Figure 3) showed that *PC1* (the most significant component) revealed a strong association of body-related factors, specifically WEIGHT, and BMI. *PC2* shows a strong correlation among variables related to the SDI. This indicates that the SDI plays a significant role in the onset of MetS, in addition to focusing on interventions related to weight and obesity management.

Figure 3 depicts the distribution of participants in clusters, where Cluster 1, highlighted in green, turned out to be the cluster most predisposed to developing MetS. The arrows emphasize the contribution of individual features to the principal components.

In the context of MetS in women, the most influential factors in *PC1* were factors related to dietary components such as sodium levels based on the dietary survey (SODIUM), SATFAT, and monounsaturated fats (MONFAT), sucrose (SUCR), and FRUCT, among others. *PC2* exhibits a trend towards variables related to poor quality of sleep and anxiety, as SLPSOB1, TRAIT_ANX, SLPNOTQ, and SLPS3 have significant values in this component. Other variables related to smoking and education (SEC_SCHOOL) also notably influence this component. This suggests that dietary control is crucial in preventing MetS among women, as well as addressing poor sleep quality and anxiety. Hence, PCA highlights relevant differences in the presentation and risk factors of MetS between men and women [88,89], which is an issue that is progressively gaining relevance in the biomedical literature [90].

The PCA results for women illustrated in Figure 4 show the distribution of participants in clusters. Similarly to the men’s analysis, the cluster most predisposed to developing MetS was Cluster 1, which is depicted by yellow dots.

### 5.4. Implications for Metabolic Syndrome Surveillance, Risk Factors, and Public Health Policy

The results of this project suggest several key findings related to the diagnosis of metabolic syndrome:Identification of known risk factors: For both men and women, specific variables were identified as strongly related to MetS. These included glucose (GLU), triglycerides (TRIG), waist circumference (WC), body mass index (BMI), and systolic blood pressure (SBP), among others. Notably, these variables are consistent with established criteria for diagnosing MetS, reflecting their importance in understanding the condition.Gender-Specific Influential Factors: This study highlights that certain factors vary in importance between men and women in predicting MetS. For instance, vitamin B12, lactose, and carbohydrates were influential in men, while total proteins, fructose, and copper were significant for women. These gender-specific variations underscore the complexity of MetS and the need for tailored diagnostic approaches. One cautionary note regarding potential outliers, specifically blood glucose and triglycerides, emphasizes their close association with the definition of MetS.Influence of Sleep and Dietary Habits: The inclusion of sleep-related variables (sleep disturbances, breathing issues) and dietary elements (cholesterol levels, nutrients) underscores their relevance in predicting MetS. These findings suggest that lifestyle factors and dietary habits are integral components in the diagnostic considerations for MetS.Potential Role of Psychological Factors: Psychological factors such as trait anxiety were included in the analysis, emphasizing the potential influence of mental health aspects in predicting MetS for both men and women.Gender-Specific Dietary Influences: For women, the analysis identified specific dietary factors like sodium levels, saturated fat, and monounsaturated fat as influential. This emphasizes the importance of considering gender-specific dietary influences in MetS diagnosis.

Understanding the gender-specific variations and influential factors highlighted in this study can inform targeted interventions that address the unique needs of both men and women. Public health policies can be crafted to recognize and address the gender-specific variations in the risk factors for MetS. By tailoring interventions to the specific needs of each gender, policymakers can enhance the effectiveness of preventive measures. Moreover, this study underscores the importance of lifestyle factors, including sleep patterns and dietary habits, in predicting MetS. Public health initiatives can thus prioritize educational campaigns and interventions promoting healthier sleep practices and balanced diets. Encouraging regular physical activity, reducing sedentary behaviors, and emphasizing the significance of maintaining a healthy weight can be integral components of public health programs aimed at preventing MetS.

Given the inclusion of psychological factors such as trait anxiety in the analysis, public health policies can integrate mental health considerations into MetS prevention strategies. Mental health awareness campaigns, stress management programs, and access to mental health resources can contribute to holistic approaches addressing the interconnectedness of mental and physical well-being. Public health campaigns can leverage the study’s findings to engage communities and raise awareness about the risk factors associated with MetS. Community-based initiatives can offer educational resources, workshops, and screenings to empower individuals to make informed lifestyle choices. By fostering a culture of health consciousness and providing accessible information, public health policies can contribute to the early detection and prevention of MetS. In view of the evolving nature of health trends and behaviors, public health policies should include mechanisms for continuous monitoring and adaptation. Regular assessments of the population’s health status, behavior patterns, and response to interventions can inform policy adjustments. This dynamic approach ensures that public health strategies remain effective and responsive to changing circumstances.

### 5.5. The Role of Social Development Dimensions in Metabolic Syndrome

It is worthwhile to recall that after applying balancing techniques, relevant associations arise between metabolic syndrome and some SDI dimensions (see Figure 5). These effects are moderate-to-medium-sized yet statistically significant. Indeed, since some of these aspects may be modifiable by public policy, it is relevant to consider them. Metabolic syndrome has been previously reported to be related to social dimensions and inequality, but also to dietary patterns [91,92]. Interestingly Soofu and coworkers [91] also report the effect that we found of an association of MetS to housing conditions and ownership of durable assets. Inadequate housing conditions, in particular, have been discussed to contribute to an increase in the risk of cardiovascular disease [93]. In fact, local residential environments may constitute significant risk factors for MetS, which is a fact that needs to be considered in order to develop environmental interventions to improve population health [94].

Restricted access to education (referred to as EDULAG in Figure 5) has also been considered a relevant feature related to MetS [95,96]. Indeed, education levels have been found to be among the best predictors of metabolic conditions in another Mexico City cohort [97]. A similar association has been reported with regards to housing (QUA_HOUS in Figure 5) [98,99]. A study in an urban Korean population found that non-apartment residents were more likely to have MetS and related phenotypes compared to apartment residents in a model that was adjusted for confounding variables such as sociodemographic characteristics, residence area, health behavior, and nutritional information awareness [93]. Sanitary conditions are known to modify both environmental conditions and even intrinsic factors such as the gut microbiota, affecting the development of MetS [100,101,102]. All of these dimensions of social development are related in a non-trivial fashion to the development of the complex pathophenotypes making up metabolic syndrome as is further evidenced by our study. However, the actual relationships between these and other risk factors remain to be investigated as open questions that must be studied in order to design targeted public health interventions.

## 6. Conclusions

In this study, logistic regression was initially utilized to identify pivotal factors linked to MetS across genders, followed by dataset balancing techniques. Our findings indicated significant variables for men, including high glucose levels, triglycerides, waist circumference, systolic blood pressure, vitamin B12, body mass index, high intake of carbohydrates, and lactose. For women, critical factors were glucose levels, triglycerides, waist circumference, body mass index, systolic blood pressure, total protein intake, fructose, cholesterol, uric acid, and copper levels. Further analysis employing SMOTE and ADASYN with RF and RPART methods re-evaluated critical features for MetS prediction in a balanced dataset. This improved model generalization by ensuring more consistent and precise training, enhancing performance, and minimizing overfitting risks. Notably, the analysis also highlighted the relevance of family history of gout as a significant factor, particularly among men. This finding underscores the potential genetic predisposition to gout, suggesting that a familial history of the condition might increase the likelihood of MetS in relatives, possibly due to shared dietary habits contributing to obesity and insulin resistance. These insights emphasize the need for gender-specific public health strategies and medical interventions, considering both the common risk factors and those unique to each gender, such as the family history of gout, to effectively manage and prevent MetS.

### Limitations

The current study has some limitations. This research was based on data from a cohort of relatively healthy adult residents of Mexico City. The regional emphasis of the study might affect generalizability; therefore, it is advisable to exercise caution when extrapolating the findings to wider populations. All data on socioeconomic status, lifestyle habits, family medical history, and macro- and micronutrient intake were self-reported. Although we trust the veracity of the information, some details may have been omitted or not remembered by the participants. Likewise, the instruments applied to evaluate physical activity, state of anxiety, and sleep quality are practical and easy to apply, but their effectiveness also depends on the truthfulness of the informants. Another limitation is our reliance on SDI data published by the Government of Mexico City, requiring trust in the data quality from this secondary source. Also, it is crucial to note that the cross-sectional design hinders causal inference, underscoring the need for future longitudinal investigations. Nevertheless, we were able to provide a comprehensive overview of the associations between metabolic syndrome, sleep disorders, the consumption of some nutrients, and contextual social development data such as quality and available space in the home, educational access, access to social security and/or medical services, durable goods access, sanitary adequacy, and electricity access. Moreover, as data balancing techniques continue to evolve, a variety of methods are emerging. However, in this study, we addressed only two of the most frequently used methods, ADASYN and SMOTE. It is important to highlight that we conducted only internal validation for our methods, emphasizing the necessity for external validation in larger populations in future studies. 

## Figures and Tables

**Figure 1 nutrients-16-00612-f001:**
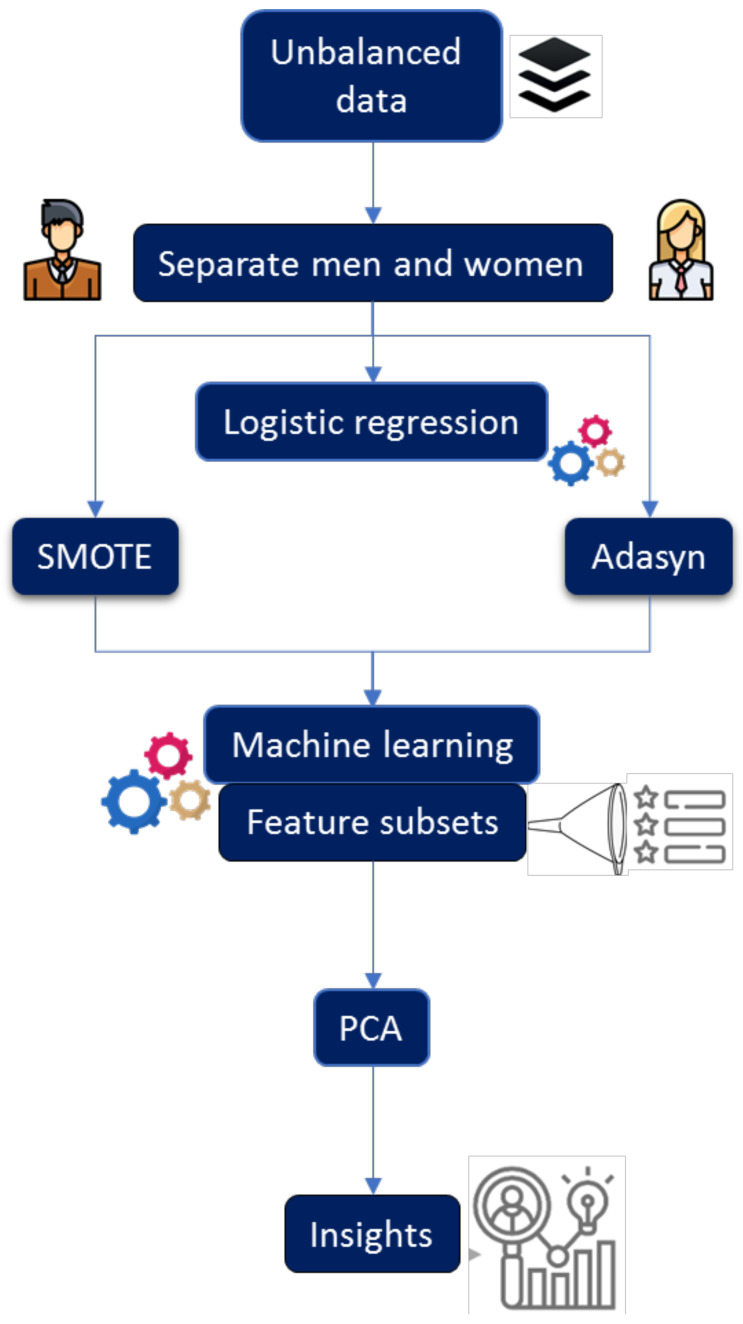
Experimental process.

**Figure 2 nutrients-16-00612-f002:**
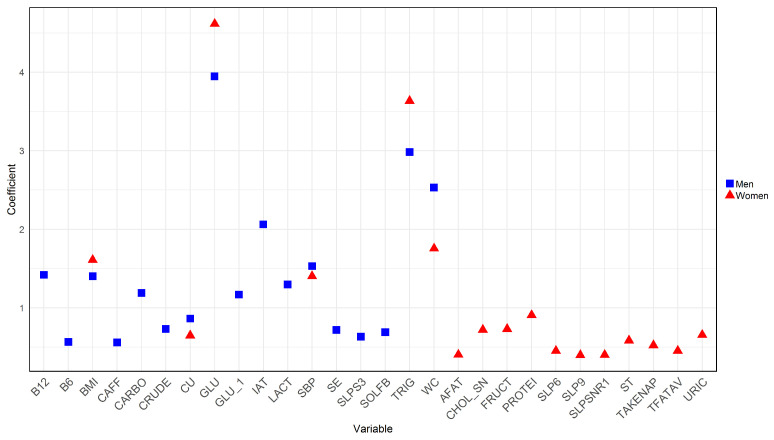
The most important variables obtained through logistic regression for men and women before data balancing.

**Figure 3 nutrients-16-00612-f003:**
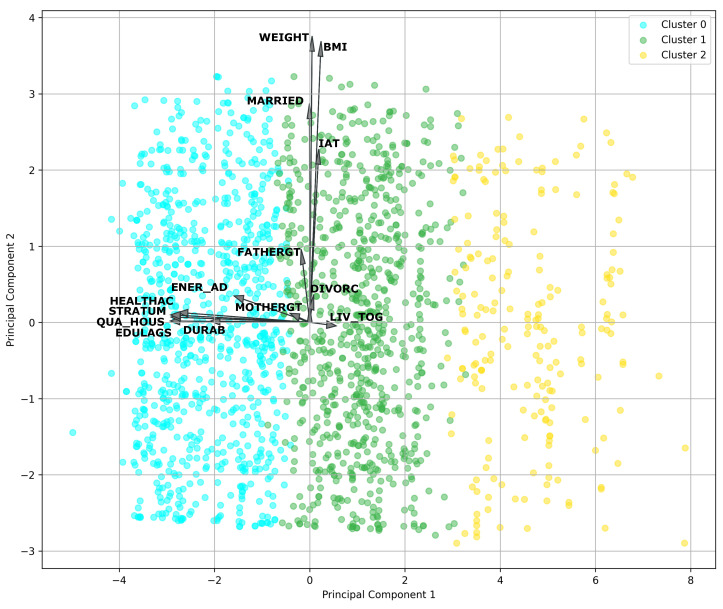
PCA of features of men for metabolic syndrome with clusters.

**Figure 4 nutrients-16-00612-f004:**
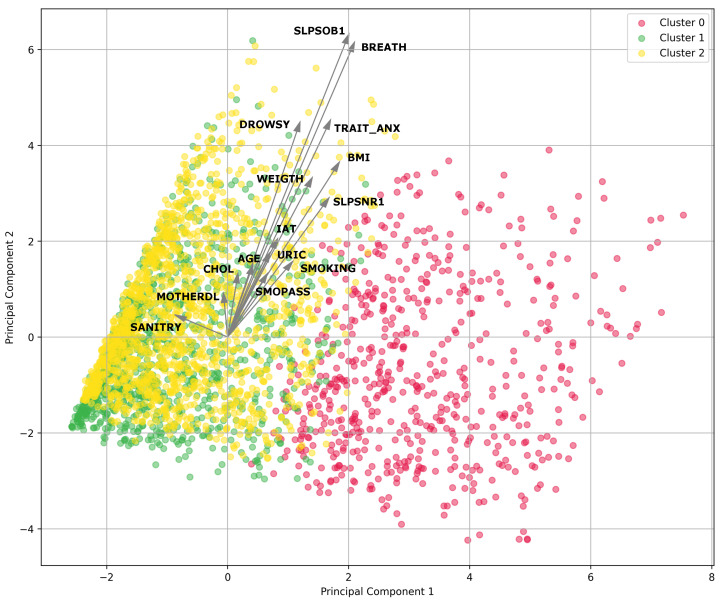
PCA of features of women for metabolic syndrome with clusters.

**Figure 5 nutrients-16-00612-f005:**
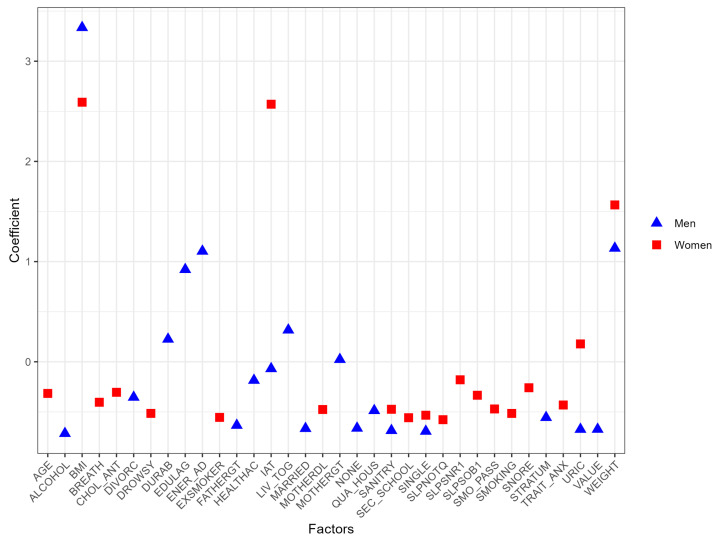
Top features for men and women considering the results of RF and RPART applying balancing techniques.

**Table 1 nutrients-16-00612-t001:** Dataset variables.

Name Variable	Description	Type
AGE	age	Continuous
WEIGHT	weight	Continuous
HEIGHT	height	Continuous
BMI	body mass index	Continuous
WC	waist	Continuous
SBP	systolic blood pressure	Continuous
DBP	diastolic blood pressure	Continuous
LIV_TOG	common-law marriage	Dichotomous
MARRIED	married	Dichotomous
SINGLE	single	Dichotomous
DIVORC	divorced	Dichotomous
VALUE	social development index by value	Continuous
STRATUM	socioeconomic stratum	Continuous
QUA_HOUS	quality and living space	Continuous
HEALTHAC	access to healthcare and social security	Continuous
EDULAG	educational lag	Continuous
DURAB	durable goods	Continuous
SANITRY	sanitary adequacy	Continuous
ENER_AD	energy efficiency	Continuous
ED_LEVEL	educational level in the neighborhood	Continuous
SEC_SCHOOL	secondary school	Dichotomous
DOCTORATE	doctorate	Dichotomous
MASTER	master	Dichotomous
SCHOOL	school	Dichotomous
BACHELORS	bachelor’s degree	Dichotomous
HIGH_SCHOOL	high school	Dichotomous
TECH_SCHOOL	technical school	Dichotomous
NONE	no academic degree	Dichotomous
TOTMET	metabolic equivalent of task	Continuous
STAT_ANX	state anxiety	Dichotomous
TRAIT_ANX	trait anxiety	Dichotomous
SLPNOTQ	sleep was not quiet	Continuous
BREATH	waking up with shortness of breath	Continuous
DROWSY	feeling drowsy or sleepy	Continuous
TROBLS	trouble falling asleep	Continuous
AWAKEN	awakens during your sleep time	Continuous
STYAWKE	trouble staying awake	Continuous
TAKENAP	takes naps of 5 min or longer	Continuous
SLPD4	sleep disturbances	Continuous
SLPSNR1	snores during sleep	Continuous
SLPSOB1	sleep short (headache)	Continuous
SLPA2	sleep adequacy	Continuous
SLPS3	somnolence	Continuous
SLPS6	sleep problems (Index I [40])	Continuous
SLPS9	sleep problems (Index II [40])	Continuous
SLPQRAW	sleep quantity	Continuous
SLPOP1	sleep quality	Dichotomous
SMOKING	smoking practice	Dichotomous
CURRENT	current smoker	Dichotomous
EXSMOKER	ex-smoker	Dichotomous
SMO_PASS	smoker passive	Dichotomous
ALCOHOL	alcohol consumption	Dichotomous
ENERGYDRK	energy drinks	Dichotomous
MOTHEROB	maternal obesity history	Dichotomous
FATHEROB	paternal obesity history	Dichotomous
MOTHERDB	maternal diabetic history	Dichotomous
FATHERDB	paternal diabetic history	Dichotomous
MOTHERHT	maternal hypertension history	Dichotomous
FATHERHT	paternal hypertension history	Dichotomous
MOTHERDL	maternal dyslipidemia history	Dichotomous
FATHERDL	paternal dyslipidemia history	Dichotomous
MOTHERGT	maternal gout history	Dichotomous
FATHERGT	paternal gout history	Dichotomous
URIC	uric acid	Continuous
CREA	creatinine	Continuous
HDLCO	high-density lipoprotein	Continuous
LDLCO	low-density lipoprotein	Continuous
GLU	blood glucose	Continuous
IAT	atherogenic index	Continuous
CHOL_ANT	cholesterol	Continuous
TRIG	triglycerides	Continuous
NA	sodium	Continuous
CALOR	energy	Continuous
PROTEI	total proteins	Continuous
APROT	proteins of animal origin	Continuous
CARBO	carbohydrates	Continuous
SUCR	sucrose	Continuous
FRUCT	fructose	Continuous
LACT	lactose	Continuous
ST	starch	Continuous
MALT	maltose	Continuous
GLU_1	glucose levels based on the dietary survey	Continuous
CRUDE	crude fiber	Continuous
SOLFB	soluble dietary fiber	Continuous
INSFB	insoluble dietary fiber	Continuous
HEMCL	hemicellulose	Continuous
CALC	calcium	Continuous
IRON	total iron	Continuous
MAGN	magnesium	Continuous
PH	phosphorus	Continuous
K	potassium	Continuous
SODIUM	sodium levels based on the dietary survey	Continuous
ZN	zinc	Continuous
CU	copper	Continuous
MN	manganese	Continuous
SE	iodine	Continuous
VITC	vitamin C	Continuous
B1	thiamine	Continuous
B2	riboflavin	Continuous
B6	vitamin B6	Continuous
B12	vitamin B12	Continuous
VITK	vitamin K	Continuous
RETINOL	retinol	Continuous
VITD	vitamin D	Continuous
VITE	vitamin E	Continuous
CHOL_SN	cholesterol levels based on the dietary survey	Continuous
ALCO	alcohol levels based on the dietary survey	Continuous
CAFF	caffeine	Continuous
AFAT	animal fat	Continuous
VFAT	vegetable fat	Continuous
TFATAV	total fat: animal + vegetable	Continuous
SATFAT	saturated fat	Continuous
MONFAT	monounsaturated fat	Continuous
POLY	polyunsaturated fat	Continuous
MS	MetS	Dichotomous

**Table 2 nutrients-16-00612-t002:** Features and values obtained through logistic regression for men and women.

Women				Men		
Variable	Coefficient	*p*_Value		Variable	Coefficient	*p*_Value
GLU	4.61438598	6.24 × 10^−59^		GLU	3.94711748	2.45 × 10^−39^
TRIG	3.63418178	1.18 × 10^−37^		TRIG	2.98165065	3.25 × 10^−24^
WC	1.75532078	2.86 × 10^−9^		WC	2.53131848	1.02 × 10^−9^
BMI	1.60919304	1.05 × 10^−6^		IAT	2.06238741	5.13 × 10^−11^
SBP	1.40299133	1.15 × 10^−12^		SBP	1.53063308	1.31 × 10^−11^
PROTEI	0.90748897	0.08529715		B12	1.41903991	0.00880359
FRUCT	0.73077934	0.23874313		BMI	1.40229014	0.00087404
CHOL_SN	0.72037259	0.06868106		LACT	1.29691863	0.00581383
URIC	0.65547784	0.01333401		CARBO	1.18935354	0.0886463
CU	0.64813271	0.17111299		GLU_1	1.1674073	0.10024746

**Table 3 nutrients-16-00612-t003:** Features of men obtained using RF with ADASYN and SMOTE applied.

ADASYN, B = 1			ADASYN, B = 5			SMOTE, K = 1			SMOTE, K = 5	
Features	Value		Features	Value		Features	Value		Features	Value
BMI	92.9499		ENER_AD	130.906694		MOTHERDL	204.657628		BMI	289.868211
WEIGHT	49.4782		BMI	104.213511		ALCOHOL	199.602686		MOTHERDL	172.071267
ENER_AD	48.8887		WEIGHT	81.5087781		BMI	198.579371		WEIGHT	169.929592
EDULAG	45.2797		EDULAG	67.7406035		SLPSOB1	111.323472		ALCOHOL	131.283664
LIV_TOG	33.3601		ALCOHOL	62.4379604		CURRENT	95.3509822		IAT	93.2909179
DURAB	31.5583		STRATUM	57.134903		BREATH	80.8262246		CHOL_ANT	63.4703128
MOTHERGT	27.5583		ED_LEVEL	55.578244		SLPD4	70.1756789		NA	49.2933568
IAT	25.7470		NONE	38.1101529		CAFF	68.9892898		CREA	45.8846962
HEALTHAC	23.4522		DURAB	36.4129389		SLP6	60.2949079		SINGLE	44.6897663
DIVORC	20.1163		VALUE	36.0130176		WEIGHT	56.9297661		SLPSNR1	35.672622
QUA_HOUS	17.4925		DIVORC	35.8243538		TOTMET	52.4806201		MOTHERDB	35.21356
STRATUM	16.1269		FATHERGT	33.7033121		ALCO	45.7609412		ENERGYDRK	34.0359073
FATHERGT	14.5872		MASTER	29.8751736		AWAKEN	39.0795326		URIC	31.8268793
NONE	14.0213		PRIMARIA	28.3852397		IAT	38.042823		AGE	27.9839119
MARRIED	13.9584		SLPSNR1	27.9671847		TROBLS	36.7528999		MARRIED	27.8864259
VALUE	13.8059		AGE	24.3706018		STYAWKE	36.2387269		DOCTORATE	24.4733499
URIC	13.7930		IAT	22.0506592		MALT	34.3472852		DIVORC	24.142464
SANITRY	13.5609		SANITRY	21.924077		BACHELORS	33.7934562		SLPOP1	23.8868609
SINGLE	13.4148		SINGLE	21.7818986		MARRIED	32.6228111		SEC_SCHOOL	22.755325
ALCOHOL	12.9798		DOCTORATE	19.8069099		SLP9	31.0845509		SLPQRAW	20.666244

**Table 4 nutrients-16-00612-t004:** Features of men obtained using RPART with ADASYN and SMOTE applied.

ADASYN, B = 1			ADASYN, B = 5			SMOTE, K = 1			SMOTE, K = 5	
Features	Value		Features	Value		Features	Value		Features	Value
LIV_TOG	447.069761		BMI	683.735277		BMI	185.940586		BMI	164.086828
BMI	402.975487		ENER_AD	619.998675		WEIGHT	131.361866		WEIGHT	132.276557
ENER_AD	338.664389		EDULAG	565.325738		FATHERGT	115.496204		IAT	131.937059
EDULAG	325.498647		ALCOHOL	355.970533		MOTHERDL	96.1708037		SINGLE	83.6531675
DURAB	285.861702		WEIGHT	295.254303		IAT	67.2839991		MOTHERDL	71.6947353
SLP6	64.2112969		DIVORC	214.489844		AGE	40.9532174		APROT	47.2274885
WEIGHT	33.1175418		NONE	200.599299		LACT	28.7681412		TFATAV	22.4867652
IAT	27.5407406		MOTHERGT	178.450647		MOTHERHT	25.3414479		ST	20.7519258
FATHEROB	14.5734264					PROTEI	14.5865884		HEALTHAC	19.7752349
SLPSNR1	13.7361635					CAFF	14.1658755		SATFAT	17.5962564
						ZN	12.4515539		HEIGHT	16.3718359
						MN	12.20696		CHOL_ANT	15.4222905
						IRON	10.5317678		MONFAT	13.9908309
						VALUE	10.2017285		CREA	13.6358167
						STYAWKE	10.1887194		URIC	11.0085972
						MONFAT	10.0410598		AGE	10.5421496
						CHOL_ANT	9.78675973		CALC	10.0034374
						ST	9.41791645		SMOKING	9.53883547
						SINGLE	9.40405705		LACT	9.34161011
						SOLFB	7.74765092		TOTMET	9.09355989

**Table 5 nutrients-16-00612-t005:** Features of women obtained using RF with ADASYN and SMOTE applied.

ADASYN, B = 1			ADASYN, B = 5			SMOTE, K = 1			SMOTE, K = 5	
Features	Value		Features	Value		Features	Value		Features	Value
BMI	208.269603		ENER_AD	344.249674		WEIGHT	321.316267		BMI	484.307061
IAT	151.849516		BMI	210.90055		IAT	294.958989		IAT	481.475021
WEIGHT	98.3094923		IAT	173.895403		BMI	253.281611		WEIGHT	339.174822
EDULAG	98.0933243		ALCOHOL	146.230976		EXSMOKER	246.78181		URIC	142.754087
LIV_TOG	82.4204188		DURAB	142.91494		MASTER	241.332636		SLPSNR1	92.0496746
ENER_AD	80.7154997		EDULAG	142.817907		FATHERDL	211.443455		CHOL_ANT	74.3706077
URIC	60.4722703		WEIGHT	128.038926		CREA	170.195583		AGE	72.769531
VALUE	53.5122927		VALUE	80.989846		MOTHERHT	125.867318		SLPSOB1	70.1959444
DURAB	48.2486067		NONE	76.4699068		SLPSOB1	125.384246		BREATH	60.3028803
QUA_HOUS	37.8080123		QUA_HOUS	62.8303545		SMO_PASS	86.2763209		TRAIT_ANX	56.4099594
SLPSNR1	31.399627		BACHELORS	56.0706757		BREATH	83.1176663		SMO_PASS	50.8288614
HEALTHAC	30.6724986		SANITRY	52.5802813		CHOL_ANT	78.8668934		SANITRY	50.3648334
SANITRY	24.2597947		HEALTHAC	45.9188536		SMOKING	57.7946015		MOTHERDL	50.0567677
ALCOHOL	24.2064626		URIC	43.8531276		TRAIT_ANX	57.3909833		DROWSY	44.564559
AGE	21.594859		SINGLE	39.5694722		SLPSNR1	51.1574483		SMOKING	44.5264858
SINGLE	18.0193809		DIVORC	37.3860944		NA	50.3156936		SINGLE	41.993735
HIGH_SCHOOL	17.1684616		AGE	33.8392029		MARRIED	48.4664641		EXSMOKER	38.9120379
SLP6	16.0530682		TECH_SCHOOL	32.4092154		SLPOP1	48.3006717		SEC_SCHOOL	38.4719692
SOLFB	14.4271683		SCHOOL	28.2955057					SLPNOTQ	35.6761924
FATHERGT	13.8839264		MARRIED	27.6425229						

**Table 6 nutrients-16-00612-t006:** Features of women obtained using RPART with ADASYN and SMOTE applied.

ADASYN, B = 1			ADASYN, B = 5			SMOTE, K = 1			SMOTE, K = 5	
Features	Value		Features	Value		Features	Value		Features	Value
BMI	664.323812		BMI	1164.1686		BMI	427.45413		IAT	483.233069
LIV_TOG	535.392713		DURAB	1117.88127		IAT	363.893488		BMI	410.367827
ENER_AD	507.53479		ENER_AD	1090.27197		SLPSNR1	259.475806		WEIGHT	409.777127
EDULAG	505.45874		EDULAG	772.049538		SLPS3	259.475806		URIC	278.65513
IAT	468.310602		ALCOHOL	655.016952		EXSMOKER	217.54026		SLPSNR1	86.0218576
			NONE	533.217568					SMOKING	31.3976405
			IAT	380.443927					SLPS3	30.5201011
			WEIGHT	366.577281					SODIUM	15.7251124
			VALUE	104.231729					ALCOHOL	12.4735987
			TECH_SCHOOL	92.1094015					SATFAT	12.1523683
									MONFAT	12.1446951
									NA	11.2712045
									VITE	10.3455105
									CHOL_ANT	9.04441276
									FATHERDB	8.09870623
									SUCR	7.16739885
									MARRIED	6.39473684
									FRUCT	4.94398493
									MALT	4.8372105

**Table 7 nutrients-16-00612-t007:** Results of the random forest models applying ADASYN and SMOTE in men and women.

Sex	Subset	Parameters	B.ACC (%)	Sensitivity (%)	Specificity (%)
Men	ADASYN, B = 1	Mtry = 9	86.22	90.93	81.50
		Ntree = 200	±0.26	±0.60	±0.41
Men	ADASYN, B = 5	Mtry = 8	85.56	87.85	83.26
		Ntree = 200	±0.34	±0.49	±0.55
Men	SMOTE, K = 1	Mtry = 10	82.86	91.51	74.21
		Ntree = 200	±1.66	±0.68	±3.45
Men	SMOTE, K = 5	Mtry = 10	75.43	90.48	60.39
		Ntree = 100	±1.29	±0.95	±2.50
Women	ADASYN, B = 1	Mtry = 10	87.12	91.10	83.15
		Ntree = 200	±0.25	±0.40	±0.29
Women	ADASYN, B = 5	Mtry = 10	86.73	88.62	84.84
		Ntree = 300	±0.20	±0.24	±0.36
Women	SMOTE, K = 1	Mtry = 10	82.55	90.48	74.62
		Ntree = 300	±0.71	±0.39	±1.46
Women	SMOTE, K = 5	Mtry = 10	88.50	91.91	85.10
		Ntree = 300	±0.40	±0.42	±0.75

**Table 8 nutrients-16-00612-t008:** Results of the RPART models applying ADASYN and SMOTE in men and women.

Sex	Subset	Parameters	B.ACC (%)	Sensitivity (%)	Specificity (%)
Men	ADASYN, B = 1	cp = 0.05	82.14	81.57	82.71
			±1.75	±3.38	±2.07
Men	ADASYN, B = 5	cp = 0.05	82.32	82.87	81.77
			±0.99	±4.67	±5.02
Men	SMOTE, K = 1	cp = 0.001	75.41	73.09	77.73
			±2.78	±4.07	±5.36
Men	SMOTE, K = 5	cp = 0.002	74.67	71.96	77.38
			±2.78	±4.07	±5.36
Women	ADASYN, B = 1	cp = 0.05	78.90	69.96	87.84
			±0.31	±0.00	±0.62
Women	ADASYN, B = 5	cp = 0.05	78.90	69.96	87.84
			±0.31	±0.00	±0.62
Women	SMOTE, K = 1	cp = 0.001	80.86	79.85	81.87
			±1.91	±3.79	±3.57
Women	SMOTE, K = 5	cp = 0.005	84.49	84.20	84.79
			±1.43	±3.01	±2.51

## Data Availability

All relevant data are contained within the article. The original contributions presented in the study are included in the article/Appendix A; further inquiries can be directed to the corresponding authors.

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
