# Peer review of "Sleep Quality, Nutrient Intake, and Social Development Index Predict Metabolic Syndrome in the Tlalpan 2020 Cohort: A Machine Learning and Synthetic Data Study"

_nutrients, 2024, doi:10.3390/nu16050612_

Round 1

Reviewer 1 Report

Comments and Suggestions for Authors

The authors outline a project to predict metabolic syndrome using data about social development, sleep quality, and nutrient intake using machine learning.  I applaud the concept for the paper since social and sleep factors are often overlooked in the context of physical health. Specific comments are below.

-You do not need to explain the structure of the paper. 

-The methods are well thought out, but the figures and tables are difficult to interpret. The manuscript would be greatly strengthened by improving the graphic depiction of the results. 

-Tables 7 & 8 runs off the page

-Why is Figure 3 in the Results section but Figures 4-6 in the Discussion section? Consider condensing into one figure.

-The discussion section reads like a rehash of the results. Where is the interpretation of what these results mean for the diagnosis of MetS? Particularly given that the title references the importance of sleep quality and social determinants but all the top variables are related to physiological measures, does this mean that future methods do not need to take sleep or social factors into account? What is the importance of the gender difference?

-The limitations section needs to actually address limitations. How does a relatively healthy adult Mexico City population affect the generalizability of the results? 

Author Response

Response to the reviewers comments

Sleep Quality, Nutrients Intake and Social Development Index predict Metabolic Syndrome in the Tlalpan 2020 Cohort: A Machine Learning and Synthetic Data study

nutrients-2841650 (before February 11th)

Reviewer 1

The authors outline a project to predict metabolic syndrome using data about social development, sleep quality, and nutrient intake using machine learning.  I applaud the concept for the paper since social and sleep factors are often overlooked in the context of physical health. Specific comments are below.

The authors are grateful to Reviewer 1 for their comprehensive review and assessment of our manuscript. We will present a point-by-point response to all your comments and concerns (to ease reading our responses appear in bold type). We are sure that by following your suggestions we were able to significantly improve the scope and clarity of our work.

-You do not need to explain the structure of the paper. 

The section explaining the structure of the paper has been removed

-The methods are well thought out, but the figures and tables are difficult to interpret. The manuscript would be greatly strengthened by improving the graphic depiction of the results. 

The graphic representation of the results has been improved.

-Tables 7 & 8 runs off the page

Tables 7 and 8 have been corrected to fit within the page margins.

-Why is Figure 3 in the Results section but Figures 4-6 in the Discussion section? Consider condensing into one figure.

We have considered removing Figures 4 and 6 and improving the legibility of the vectors and labels in Figures 3 and 5. Figures 4 and 6 were initially included as support to better visualize the direction and labels of the PCA feature vectors in Figures 3 and 5.

-The discussion section reads like a rehash of the results. Where is the interpretation of what these results mean for the diagnosis of MetS? Particularly given that the title references the importance of sleep quality and social determinants but all the top variables are related to physiological measures, does this mean that future methods do not need to take sleep or social factors into account? What is the importance of the gender difference?

Thank you for pointing this out! We have considered these issues in the revised version of the Discussion. In particular, we have included a new subsection in the discussion related to the implications of our work for metabolic syndrome surveillance, risk factors and public health policy. Where we discuss how having these types of predictors may serve to get early warnings, identify novel risk factors and provide public health policy officials with additional tools to design prevention and intervention programs. 

-The limitations section needs to actually address limitations. How does a relatively healthy adult Mexico City population affect the generalizability of the results? 

We have added a subsection in the Conclusion section to briefly summarize the limitations of our study including the issue of generalizability given our study population design.

Reviewer 2 Report

Comments and Suggestions for Authors

This is an interesting research article with adequate novelty and a study design of high quality. Some points should be addressed.

- In the Abstract, the introduction is quite long. The authors should try to decrease the length of the introduction of the abstract. They should simultaneously add more details about their results of their study, including also at the end of the abstract a conclusion statement and their future perspectives in the topic of their research.

- In the introduction section, the 2nd paragraph (lines 56 - 69) and the 7th paragraph (lines 111 - 121) include several repetitions and should be merged into one paragraph.

- In the sentence in lines 75 - 77, something is missing ant this sentcence need revision for its syntax. Maybe, it could revised as "Evidence suggests a close association between 75 SDI and sleep disturbances, which may be influenced by social and economic factors, such as income and education". 

- In the introduction section, some statement should be included concerning the well-recognized screening tools, which were used to assess sleep quality. (mayby in the 1st or 4th/5th paragraphs).

- The Methods section is very well-organized and well-written, including all the appropriate information in details. Thus a study design of high quality and validity characterized this study.

- The resolution of Figure 2 should be improved a bit.

- Again, Figure 3 needs improvement concerning its resolution (e.g., the word into the figure are not quite visible).

- In line 336, something is missing, e.g. : "...(see figure ??)...".

- In line 339, the authors used both "Furtermore" and "also" in the same sentence. Please use only one of them. This is also happened in the following sentence by using both "Similarly" and "also". Please use only one of them.

- Figures 4 and 5 need an improvement concerning mainly the size of the words included into the figures.

- Please, check that each abbreviation is explained only one time, when it is reported for the first time.

- At the end of the Discussion section the authors should thoroughly report the strengths and the limitations (e.g. what about confounding factors? Could the results reproduce by using another sleep quality assessment screening tool like PSQI?, et al.) of their studies.

- The Conclusion is very confusing and not easily inderstood for the readers. The authors should simply report their main findings without using the abbreviations.

-Moderate English language editing is recommended by revising misspelings, syntax/grammar and typos errors.

Comments on the Quality of English Language

Moderate editing of English language required

Author Response

Response to the reviewers comments

Sleep Quality, Nutrients Intake and Social Development Index predict Metabolic Syndrome in the Tlalpan 2020 Cohort: A Machine Learning and Synthetic Data study

nutrients-2841650 (before February 11th)

Reviewer 2

This is an interesting research article with adequate novelty and a study design of high quality. Some points should be addressed.

We are thankful to Reviewer 2 for their scholarly discussion and evaluation of our manuscript. In what follows, we will present a point-by-point response to all your comments and concerns (our responses appear in bold type to ease reading). We are confident that by following your suggestions we will be able to significantly improve the scope and clarity of our work.

- In the Abstract, the introduction is quite long. The authors should try to decrease the length of the introduction of the abstract. They should simultaneously add more details about their results of their study, including also at the end of the abstract a conclusion statement and their future perspectives in the topic of their research.

Following your suggestions, we have modified the abstract by simplifying the introduction, expanding the details in the results section, highlighting the relevance of the study, and indicating directions for future research.

- In the introduction section, the 2nd paragraph (lines 56 - 69) and the 7th paragraph (lines 111 - 121) include several repetitions and should be merged into one paragraph.

Thanks for the recommendation, we have merged these ideas into one paragraph.

- In the sentence in lines 75 - 77, something is missing ant this sentence need revision for its syntax. Maybe, it could revised as "Evidence suggests a close association between 75 SDI and sleep disturbances, which may be influenced by social and economic factors, such as income and education". 

We have refined the text to improve clarity, highlighting the compelling evidence for a significant link between the Social Development Index (SDI) and sleep disturbances. This link is profoundly shaped by socioeconomic factors, including income and education levels, which affect both healthcare access and lifestyle choices critical to optimal sleep quality.

- In the introduction section, some statement should be included concerning the well-recognized screening tools, which were used to assess sleep quality. (maybe in the 1st or 4th/5th paragraphs).

A statement was added at the end of the first paragraph in the introduction of the document, highlighting the use of the Medical Outcomes Study Sleep Scale (MOS) as an important tool for assessing sleep quality and its influence on health.

- The Methods section is very well-organized and well-written, including all the appropriate information in details. Thus a study design of high quality and validity characterized this study.

Thank you! We have also improved the figures in this section.

- The resolution of Figure 2 should be improved a bit.

The resolution of Figure 2 has been enhanced

- Again, Figure 3 needs improvement concerning its resolution (e.g., the word into the figure are not quite visible).

The resolution has been enhanced, ensuring clarity and visibility of the text within the figure. 

- In line 336, something is missing, e.g. : "...(see figure ??)...".

The figure number has now been specified. 

- In line 339, the authors used both "Furtermore" and "also" in the same sentence. Please use only one of them. This is also happened in the following sentence by using both "Similarly" and "also". Please use only one of them.

These redundancies have been corrected.

- Figures 4 and 5 need an improvement concerning mainly the size of the words included into the figures.

Figure 4 has been removed following improvements in the clarity of vector labels in Figure 3. Similarly, the legibility of labels in Figure 5 has been enhanced. We considered removing Figures 4 and 6 with the aim of condensing the information in Figures 3 and 5. Initially, Figures 4 and 6 were included to facilitate a clearer visualization of the direction and labels of the feature vectors in the principal component analysis depicted in Figures 3 and 5.

- Please, check that each abbreviation is explained only one time, when it is reported for the first time.

The observation has been corrected. Each abbreviation is now explained the first time it is mentioned

- At the end of the Discussion section the authors should thoroughly report the strengths and the limitations (e.g. what about confounding factors? Could the results reproduce by using another sleep quality assessment screening tool like PSQI?, et al.) of their studies.

A limitations subsection has been in the revised manuscript.

- The Conclusion is very confusing and not easily understood for the readers. The authors should simply report their main findings without using the abbreviations.

In response to the observation about the clarity of the conclusions, we have revised them to articulate our main findings more clearly and without such excessive use of abbreviations. 

-Moderate English language editing is recommended by revising misspelings, syntax/grammar and typos errors.

Syntax and grammar of the whole manuscript has been double checked.

Round 2

Reviewer 2 Report

Comments and Suggestions for Authors

The auhtors have now improved significantly their manuscript.